# Research and the Challenge of Participation—The Experience of the Research Project "Distressed Neighborhoods through the Prism of Youth" (ANR Pop-Part)

**Jeanne Demoulin** [1,2,*] **and Marie-Hélène Bacqué** [2]

1 Centre de Recherches en. Éducation et Formation (CREF), Paris Nanterre University, 200 Av. de la République, 92000 Nanterre, France
2 UMR LAVUE, Paris Nanterre University, 200 Av. de la République, 92000 Nanterre, France; mbacque@parisnanterre.fr
* Correspondence: demoulin.j@parisnanterre.fr

**Abstract:** Based on a participatory research project on the practices and representations of young people in distressed neighborhoods, this article examines the contributions and limitations of a participatory approach in terms of scientific production. How does participation affect social science research? How does it challenge methodological and epistemological principles, knowledge building, and the nature of that knowledge? To what extent is it heuristically stimulating from this point of view?

**Keywords:** participatory research; youth; working-class; distressed neighborhoods

## 1. Introduction

Although it has existed in France for years, so-called participatory research, also known as cooperative, partner, and action research (Juan 2021), is now attracting renewed interest. While participatory research claims to adhere to different approaches, currents, and objectives, it has one common goal: "to change the standard process of knowledge production, by involving people or groups who do not usually have this role" (Deroubaix and De Coninck 2013). It can be instigated by researchers (Le Méner and Oppenchaim 2013; Purenne 2018) or by civil society stakeholders, including national actors, such as federations of voluntary organizations, local actors (Demoulin and Charleux 2022), and social or community movements. Based on the participatory research conducted by POP-PART on the practices and representations of youth in distressed neighborhoods, this article examines how the participatory methodology contributes to scientific production and the limits of this methology. POP-PART's research was conducted from 2017 to 2021 and involved 15 researchers, 15 civil society leaders and professionals working with young people, and 120 young people from ten working-class neighborhoods in the Paris region. The research was based on workshops, which were organized with young people in each neighborhood. It involved producing and discussing videos, mental maps and texts, individual interviews, and later, meetings to bring together all the research participants. This last phase was extended, with time allocated to co-writing a joint publication. Both authors of this article are coordinators of POP-PART's research. Although the analyses developed here draw on the research team's reflections, they reflect the authors' own opinions and experiences. One of the authors is a teacher-researcher in educational science, with a background in urban planning. The other is a sociologist and urban planner. They have both worked on participatory processes in working-class neighborhoods.

Participatory methodologies are not new. They have developed significantly in recent years in natural and environmental sciences and the health sector (Houllier et al. 2017). In social sciences, they are applied by researchers in educational science or social work in

most cases. Today, they are being institutionalized, as shown, for example, by the increase in government funding, seminars, and conferences on the topic.

In other contexts, such as Canada, the United States, or Brazil, these research currents have greater recognition and are better structured than in France. They have their own journals and professional organizations. This discrepancy arguably reflects the impact that national or regional differences may have in terms of the history of the development of social sciences, the conception of knowledge, and the relationship between university and society (Rubião 2011). In France, it can partly be explained by the influence of Emile Durkheim's theory of sociology and later by Pierre Bourdieu's theory, in which the epistemological break is made central to the researchers' stance (Burawoy 2019).

Furthermore, even when participatory or partner research is an integral part of scientific institutions (Tremblay and Demers 2018), such as in Quebec, there is little dialogue between different academic or participatory research practices (Dumais 2011). Many studies now use the typology proposed by Michael Burawoy, which differentiates between academic, public, and expert research. Yet, how these different approaches to producing sociological knowledge influence each other and interact is seldom considered. Public sociology or participatory research only appears to be legitimate when it relates to action and the political sphere; there is no acknowledgement of its methodological or conceptual contributions.

In this article, we draw on POP-PART's research with young people and professionals to investigate how participation affects research in social science. We strive to go beyond the principled positions of participatory research. Our approach is consistent with other studies that seek to open the "black box" to examine how this kind of research is produced, the questions it raises at each stage of the research process, and the trade-offs involved. How does it challenge methodological and epistemological principles, knowledge building, and the nature of that knowledge? What makes this heuristic approach exciting?

First, we will revisit the body of theoretical research, which endorses these research practices and has influenced our own work. In the specific case of our research, how is this methodology expected to help us understand distressed neighborhoods and their youth? Thinking in terms of youth experience implies methodological and ethical choices, some of which emerged during the research process. We will address this issue in a second phase, before examining the scientific challenges raised by participatory research and its limitations in the light of this experience. Here, we will consider participatory research from the researchers' point of view, rather than in terms of how it affects other participants, which could be a subject for future discussion. Presenting our research results is not the object of this paper, although we discuss some findings to illustrate our point. For a panorama of the research findings, which are hard to sum up in a few lines, readers can consult the joint publication that resulted from the research (Bacqué and Demoulin 2021), as well as the companion website: https://jeunesdequartier.fr (accessed on 12 July 2023) where the scientific articles published by the POP-PART team can be consulted.

## 2. Why Participatory Research?

In given social circumstances, the emergence of participatory research raises issues that are specific to social science disciplines in terms of their construction and their relationships to society. For analytical purposes, we identify three interlinked issues, which are upheld by participatory methodologies and guided our approach: democracy and the relation to action, epistemology, and empowerment. We will use the general term "participatory research" to talk about research that seeks to work "with", "jointly", in "partnership" (Soulière and Fontan 2018) with non-scientific actors. For further discussion on terminology, there are several references on the subject (Cornwall and Jewkes 1995; Anadon 2007; Gillet and Tremblay 2017).

### 2.1. Science in Society and Relations to Action

A preliminary set of concerns includes distinct but interlinked questions about democratizing research, its relation to society, its societal relevance, and the relationship between knowledge and action. In this context, non-researchers are mobilized to take part in scientific production in various ways. For example, these actors are involved in selecting the subjects for analysis, defining protocols, interpreting and even conducting the survey, or they are simply shown the results to determine possible actions. Action research and partner research are often used in fields of study pertaining to health, education, community living, or international development. This is illustrated by the significant body of literature, especially anglophone and Quebecois. Action research aims to involve professionals and users, by integrating their professional and/or user know-how to improve the organization and management of public services.

Participatory research describes various forms and dynamics depending on whether it is initiated by researchers or social stakeholders, concerns social change or management, offers support for social movements, or is a way to fund research. Andrea Cornwall and Rachel Jewkes have shown the limitations and possible instrumentalization of the so-called participative action research (PAR) practices which are promoted by international development institutions (Cornwall and Jewkes 1995). The necessity of "social commitment", geared to identifying goals of social value for research and universities, is gradually becoming part of university' strategies and how they relate to their territories. It is a positive asset and gives them a competitive advantage (Vergnaud 2018). At the other end of the spectrum, participatory research is used in sociology, which is committed to dominated actors, in order to shed light on social movements and fuel their actions (Nicolas-Le Strat 2018; Bénit-Gbaffou et al. 2019). The South African sociologist, Karl Von Holdt, suggests substituting the notion of public sociology, as defined by Burawoy (Burawoy 2009), with critical engagement in sociology. This notion clarifies the specificity of a knowledge production process, in which there is tension between the political sphere and the sociological sphere. It highlights the conceptual and methodological innovations that emerge and, according to the author, it will help transform sociology overall (Von Holdt 2020).

### 2.2. Epistemological Issues

Participatory practices hinge on the recognition of different types of knowledge (academic, action-, and experience-based). They invite us to discuss these knowledge types and to articulate them in the knowledge production process. This challenges the status of scientific knowledge and how it is built. Some researchers, for example, question whether involving practitioners in the research process has added value and whether their contribution to "a greater scientific truth" is valid (Bussières and Fontan 2011). Above all, it could undermine scientific objectivity, since "the genuine researcher is the one who can suspend their judgement about the social realities they are studying, adopt a neutral position [with regard to] their subject of research" (Bussières and Fontan 2011). The research current featuring "research with" has been the subject of many publications in recent years, particularly Quebecois. It questions the dialogue between scientific knowledge and "common knowledge" (Bessaoud-Alonso 2017). It also examines the difficulty of holding together "the application of deeper citizen participation, on the one hand, and the production of research that meets the scientific requirements of quality and temporality, on the other" (Espinosa 2020).

Since the 1980s, feminist research has examined the question of objectivity. After realizing that women were absent in scientific production, feminist researchers called for the inclusion of their knowledge, i.e., "situated knowledge", which was built from their experience, their material life, and their dominated condition (Hartsock [1983] 1987; Collins 1986). Thus, the construction of feminist research developed from the observation that scientific studies were not neutral, but based on social stereotypes of gender and race. The philosopher, Sandra Harding, puts forward the idea of "strong objectivity" (Harding 1992), as opposed to the "weak objectivity" of sciences, which claim to be neutral but

have overlooked half of humanity for decades. In contrast, objectivity would improve if the context in which it is produced, and researchers' commitment were considered as potential resources. Building scientific knowledge would then be based on diversifying analytical supports, establishing a dialogue between contradictory interpretations, and using conflicts as a heuristic tool. This proposal triggered considerable debate in feminist research. It overturns the principle of the epistemological rupture and challenges the notion of necessary distance regarding commitment and the political sphere (Bourdieu 1997). Feminist research actually emerged from the experience of political struggle and it assumes this filiation. Therefore, research is seen as contributing to social change, not merely because it offers insight to agents with regard to their dominated state and the structures of inequality, but because it is fueled by their struggles and subjectivity. These reflections have nourished our methodology.

### 2.3. Participatory Research and Empowerment

The epistemological question raised by the feminists encountered a series of studies on subalterns (Spivak 2009), the dominated or lower classes, along with reflective and political concerns related to the decolonialization of knowledge (Smith 1999, p. 15). Therefore, the challenge is to counterbalance the researchers' positions of domination in terms of the production of legitimate representations and discourses; refusing to "endorse" the fact that researchers, who are "demographically homogenous [and] far removed from conditions of oppression [...] study and develop policies for others and confuse the products and sources of oppression"; and to reject the resulting epistemological violence (Fine 2013, p. 695, author's translation). The notion of "critical community-engaged scholarship", put forward by Cynthia Gordon da Cruz (2017), specifically addresses this issue by providing principles "for establishing university–community coalitions that work to dismantle structural inequities in our democracy" (p. 381).

This issue may lead to "right to the research" (Appadurai 2006). In Latin America, during the 1970s, liberation theology and neo-Marxist approaches to development fueled a "popular science" movement (Fals-Borda and Rahman 1991; Gonzalez-Laporte 2014). The "pedagogyof the oppressed", propounded by the philosopher and teacher, Paolo Freire (Freire 1974), is thus based on a dialogue between intellectuals and the oppressed, and between action and reflection. It fits into a framework of empowerment and challenges power relationships.

While we have sought to fit our research into this framework, this article does not develop the scope of our research with regard to empowerment. Nonetheless, we have taken Haraway's double warning seriously. "The positionings of the subjugated are not exempt form re-examination, decoding, deconstruction, end interpretation; that is, from both semiological and hermeneutic modes of critical enquiry. The standpoints of the subjugated are not 'innocent' positions. On the contrary they are preferred because in principle they are at least likely to allow denial of the critical and interpretative core of the knowledge ( . . . ) 'Subjugated' standpoints are preferred because they seem to promise more adequate sustained, objective transforming accounts of the world. But how to see from below is a problem requiring at least as much skill with bodies and language, with the mediations of vision, as the 'highest' techno-scientific visualization" (Haraway 1991, p. 191).

"Learning to see from below" involves keeping together an objectifying posture and a participative posture. As Jean-Louis Genard and Marta Toca I Escoda argue, "these two postures are not totally opposed, particularly because the 'objectifying' posture can be considered as a possible stage in the 'participatory' posture" (Genard and Roca i Escoda 2010). However, "they draw specific ethical requirements that the sociologist will inevitably have to negotiate" (p. 15). The requirements linked to methods of scientific validation, which the researcher strives to meet, may conflict with the forms of trust, reciprocity and control implied by the participative ethos (Payet et al. 2010). This involves negotiation in

terms of methodology, data collection methods, and how data are interpreted and used. How was this negotiation for us? What specific issues does it highlight?

### 3. "Learning to See from Below"

Researchers who study working-class and distressed neighborhoods are up against a dual stumbling block: fascination and idealization on the one hand, and a normative and condescending view on the other (Grignon and Passeron 1989). Approaching distressed areas through youth exacerbates these difficulties, while at the same time it justifies the choice of this research topic. Youth are the figureheads of the stigmatization of working-class areas. With the emergence and consolidation of the category "jeunes de quartier" in France, youth living in distressed neighborhoods are equated to a deviant and problematic population. The "youth" categorized in this way are the focus of public debate, in which they rarely have a voice. Indeed, they suffer from a form of social, cultural, and political dispossession. How can we study this complex field without being part of this process and without adopting a naïve posture. How can we challenge this categorization and, simultaneously, apply it?

Acknowledging these tensions appeared to be possible with the participatory approach. While it could not resolve them, it made them the focus of the research. "Youth have traditionally had no control over inquiry processes and outcomes of social science research, even when they were the focus of the work. This powerlessness reflects hierarchical social power relations and widespread assumptions that young people lack the capacity to fully understand their experiences and effectively address their own needs". (Rodríguez and Brown 2009) (p. 6). However, some studies have attempted to address this adult-centered generational asymmetry (Caron 2018; Gaudet 2020), which is coupled with social asymmetry. Thus, the inclusion and active participation of youth or teenagers in the research approach is considered as a criterion for scientific validity. Approaches involving youth start by "recognizing their specific knowledge, their views on themselves and their universe, others, the society and the world" (Soulière and Caron 2017). Our research fits into this framework.

Our aim was twofold: (1) to try to propose "a more adequate, sustained objective transforming account of the world" (Haraway 1991, p. 1991), which would make it possible to overcome certain caricatural hegemonic representations; (2) to consider and work on the relations of domination in knowledge production, using a reflective and participatory approach to science. The focus of our research was production by and with young people. The goal is scientific because it involves producing and discussing knowledge based on young people's experience. It is also political because it allows a social group to shape how it is represented in the public sphere. In so doing, we were aware of the danger of essentialism with regard to "jeunes de quartier", a category that we wanted to challenge. Hence, we broadened the scope of our research from a socio-urban point of view and worked with youth with diverse socio-economic backgrounds and family trajectories. We focused on their individual and group production, so that experiences and different opinions could be expressed and discussed. This diversity was one of the focal points of our analysis.

In order "to learn to see from below", we sought to establish the conditions for expression, listening, discussion, and then writing, by reducing the asymmetry of the partnership as far as possible. This involved methodological openness, which was inspired by participatory practices and "éducation populaire", which promotes critical thinking and emancipation. It is not opposed to the classic practices found in sociology, but is complementary, involving a step-by-step re-examination of the conditions for detachment, criticism, and reflexivity.

#### 3.1. Building the Least Asymmetric Partnerships

Many studies have pointed out the asymmetry regarding the positions of researchers and their partners in participatory research (Cornwall and Jewkes 1995; Gaventa and

Cornwall 2008). The hierarchy of knowledge and social positions cannot actually be erased simply because participants want it to go away. Nonetheless, we tried to limit it by building a partnership that includes researchers from different disciplines, members of civil society organizations, local authorities working in disadvantaged neighborhoods, and the young people who live there.

We included youth practitioners in our research team for several reasons. First, because an earlier participatory research project (Mapcollab research conducted in partnership with the INRS Montreal) highlighted the void and subsequent frustrations caused by the withdrawal of researchers once their work was over. In our view, this raised an ethical question about the researchers' responsibility. Therefore, we sought to create the conditions for possible continuity. Second, the professionals working in the field have specific knowledge and have already formed a relationship with the young people. While this is an advantage, it may generate bias. In several cases, our research prolonged earlier joint projects, which meant we were able to benefit from long-term trust.

The professionals helped form the youth groups. We sought to respect diversity in terms of origins, trajectories, social situations, schooling, and by mixing girls and boys. We worked with ordinary youth, who are not the most disadvantaged but remain largely unseen. This profile corresponds to our research targeting, but can also be explained by the fact that it was not easy involving the most disadvantaged youth. The research was conducted in ten cities in Île-de-France: Aubervilliers, Clichy-sous-Bois, Corbeil-Essonnes, Nanterre, Pantin, Paris XVIII, Saint-Denis, Suresnes, Vert-Saint-Denis, and Villeneuve-la-Garenne. Approximately 120 young people, aged from 15 to 33, took part. The group had an equal number of girls and boys (35% were aged between 19 and 22). The large majority were born in France (86%) and had at least one immigrant parent (88%) from very diverse origins: the majority come from the Maghreb and sub-Saharan Africa. Other origins included: Egypt, Turkey, Chile, Russia, Serbia, Montenegro, and the Caribbean. Almost 86% claimed to have a religion, of which over 85% claimed to be Muslim. Most of their parents were workers or employees with little or no qualifications. Lastly, over two-thirds (67%) were high school students or undergraduates, while the young working people also included those in and seeking employment. Several of the young people were the first in their family to continue studying beyond high school.

To limit the asymmetries in the group research project, we started by identifying forms of recognition for everyone's work. We decided to compensate the civil society organizations and the young people. A remuneration of EUR 500 was planned for each young person to attend workshops organized in the area, to make a video, a semi-structured interview with a researcher, and to take part in the two-day metropolitan workshop to bring together all the young people involved in the research. Compensation for the latter triggered debate between the professionals and researchers. Some youth practitioners, used to organizing projects with young people, were concerned that the young people, who already have consumerist tendencies, would develop a mercantile approach to the activities proposed by their organization. This sparked the first tension between the educational goal and the research goal. Some researchers also expressed reticence, either because they were assuming the role of educator or because a monetarized relationship seemed contrary to their ethics and might undermine the sincerity of the exchange. The team of researchers/professionals managed to reach an agreement. In most areas, the young people were only told about the compensation after they had been recruited. An intermediary solution was found in two areas, where professionals strongly encouraged the young people to pool their pay in order to organize a trip overseas. These trips extended the research framework because the researchers took part and were able to use the opportunity for observation, discussion, and group analysis.

In most cases, the young people welcomed the payment as a pleasant surprise, rather than a motivating factor, at least when the research began. However, as the payment required attending all the workshops, it was an incentive to complete the project. It also caused some tension when the young people had to wait several months before being paid.

However, the main determining factor in youth participation was undoubtedly the feeling that their knowledge and their words had gained recognition. It was a good surprise that they asked if they could continue with the meetings beyond what was planned in the initial commitment. For example, 80 youths attended a weekend of writing workshops, which was organized at their behest. The symbolic value of cooperating with university researchers is significant for the young people, many of whom experienced bumpy educational trajectories and had a complicated relationship with school.

*3.2. Experimenting with Participatory Tools*

We adopted tools that were new to part of the team in order to "learn to see from below", to take into account the plethora of narratives and to mobilize the analytical capacities of our partners, individually and collectively. This multiplied the forms of expression.

The first stage of research took the form of workshops, which were organized in each of the neighborhoods over a 6-month period. We drew inspiration, in particular, from the traditional social science methods, such as focus groups. We also used participatory methods of facilitation, which some of us teach and/or have studied and deconstructed during our research. We chose techniques combining narratives with individual and group reflection, which allow everyone to speak, express their view, and describe an experience. Having fun was a priority to help the young people concentrate and to avoid reproducing a school environment. "Education populaire" methods proved very useful to encourage participants to adopt reflective postures, to raise questions about individual and shared histories, and to try to understand and explain them. These workshops were organized around the neighborhood experience, but the aim was also to discuss youth practices and representations at other scales.

During these workshops, researchers took it in turns to adopt different positions so they could observe, facilitate, but also interact in the discussion. We decided to work in pairs to alternate between the roles of observer and facilitator. Respecting this clear distinction was not always easy and the researchers sometimes found it tricky to change their stance. The curiosity that the young people expressed about us served as a tool for dialogue and helped to re-establish a balance in the discussion. During a workshop about the link between personal histories and history, for example, the researchers and professionals working with young people also shared historical and personal events that were important for them. During non-mixed group discussions about the relationships between girls/boys, several researchers answered the girls' questions about what they had experienced at the same age. In general, we decided that once the young people had built their own analysis, we would not remain outside the discussion, but would be open, would express our views when asked, and answer the questions raised.

In this way, during the workshops and meetings, a knowledge production process developed and individual and collective viewpoints emerged.

In Pantin, for example, the first workshops focusing on the urban experience led young people to build a personal discourse, as well as a group position about the urban changes in their city center. Going back and forth between the individual (the words they chose to define their city, their photographs of the city, etc.), and the group (discussions about the words and photographs), revealed how ambivalent their respective positions were regarding the current process of gentrification (which is reflected in the arrival of a more affluent population and an increase in property prices): a deep-rooted fear about being excluded from their city is juxtaposed with their enthusiasm about seeing their city embellished, seeing the image of the region and its inhabitants improve, and seeing amenities flourish, which they can enjoy in their leisure time. The same ambivalence was manifest in Paris in the 18th arrondissement, where the presence of migrants in the public space adds an element of tension. These shared reflections allowed us to go beyond the classic approaches to gentrification, which focus on those responsible for gentrification.

In Nanterre, in the Petit-Nanterre neighborhood, work on family backgrounds was organized in several phases, which also linked the individual and the group: family trees

were produced individually; family figures were presented to the group; and differences and similarities were identified. As a result, the young people reconstructed their family histories and proposed a personal analysis of the trajectories of some family members. Thus, the young people revealed how similar their family histories were. They saw themselves as part of a whole by showing how closely linked their personal histories were to the history of Algerian immigration in France and, by extension, to the history of their neighborhood, an area marked by slums and the large-scale construction of social housing units.

The non-mixed workshops addressing the issue of the relationships between girls/boys were extremely productive. In this space, many words were expressed about questions of gender relations, particularly linked to the role of religion in everyone's life history. These workshops raised awareness and led to questions that could be explored together, as told by a young participant: "the discussion with the girls was one of my favorite moments. [...] We noticed with [another girl] that we didn't have the same background even if we lived . . . The experience of two black Muslim girls, who live in the same town, isn't the same".

The video was used as a means of expression and as an analytical tool during the workshops. In this way, our method was in line with research that uses a participatory audiovisual approach. The methodologies are very diverse, but video is upheld as a means to ensure that everyone's voice is heard because it maximizes "opportunities for participation" (Hart 1992). After a training session on filming and how to use the equipment, we proposed that the young people make video clips of two or three minutes on their smartphone or digital tablets. On this occasion, the team was extended to include specialists to help the young people realize their projects, while giving them the freedom to be imaginative and creative.

Similar to other researchers who have used video (Soulière and Caron 2017), we can confirm that the young people were excited about making the films and enjoyed the project. They were proud of the finished product and were proud to present it to other young people. Some chose fiction, others chose a guided tour or interviews. Building scenarios, filming (often in groups), and screening the pictures triggered numerous improvised discussions. Some young people, especially girls, who seldom expressed their opinion during the debates, used it as a means of expression and, thus, brought up new topics for debate. Making a film helps develop an analytical approach and gives research access to "the world of youth in all its complexity, in different places and beyond the workshop room" (Soulière and Caron 2017).

One of the difficulties encountered by the researchers was to enter into these representations and find the analytical tools to work on the material. Some were disappointed by the "quality" of the images or because they did not see an explicit discourse. They regretted that many videos only highlighted "positive" points and minimalized the "negative" points or made them invisible, and that they lacked the "richness" of the workshop discussions. Yet, the video project gave the young people a means of expression and a basis for analytical construction at different stages, which included making the videos, selecting the topics, and presenting and discussing them. They brought to light problems that had not been discussed in the first workshops. This material is meaningful for the work as a whole and requires specific methods of analysis.

Thus, during the first workshop in the 18th arrondissement, several young people declared that there were too many blacks and Arabs in the neighborhood, in reference to the refugees in the public space. All these young people belong to racial minorities and discussion was difficult. Two girls, who struggled to assume a position in the debate, chose this subject for their video and interviewed two refugees. The video concludes with the words: "you see, they aren't . . . " In the same neighborhood, a group of girls invents a small scene where they show how boys try to impose limits in the neighborhood and control the girls' movements between the 18th and 19th arrondissements. In this way, they shed new light on topics essentially discussed by boys in the workshop linked to issues about fighting, which seldom involved the girls.

The second research step involved a change in scale and a more general analytical approach: youths from the ten neighborhoods attended a weekend event to compare and contrast their output and to deepen the analysis, by considering different situations. We were surprised by their tremendous participation (although it was part of the financial commitment), by their expectations, and by the energy they invested in the meetings. There were video presentations to youths from other neighborhoods and group discussions on topics chosen by the young people. This generated a group dynamic, allowed the young people to gauge how their analyses had developed, and to grasp the contributions that this might represent. The heuristic dimension of comparison and alterity was demonstrated by Michele Fines in studies with youths in New York. Here, it also developed a productive analytical base, which revealed criteria of socio-spatial hierarchy, common markers or, on the contrary, specific features between neighborhoods.

In response to the young people's request for another meeting between neighborhoods, a weekend dedicated to writing was held several months later. More than half of them attended although there was no compensation this time. Over the weekend, and at occasional meetings that followed, where specific subjects were discussed in small groups, the young people produced individual and group texts in view of a joint publication on topics, such as culture, discrimination, sport, the media, girl/boy relationships, and violence. Here, they took the initiative to enhance the initial methodological protocol and extend the group work, which illustrates their ownership of the research process.

### 3.3. Keeping a Critical Distance, Hybridizing Methodological Approaches

The use of participatory methods and tools encouraged the young people to express themselves individually and in groups and to discuss with researchers. Nonetheless, sociology's more traditional apparatus was not replaced entirely. During the workshops, for example, we used mental maps and guided tours, and tried involving the young people in the analysis of the elements collected.

The researchers conducted individual interviews after the workshops, focusing on topics that had emerged. It gave them the opportunity to assess the approach adopted. This "regulated improvisation" is a well-known exercise (Bourdieu et al. 1968), where the researcher finds themselves facing the interviewee. It has been the subject of numerous discussions and methodological guides, but here the rules and conditions were transformed because of the work performed with the young people beforehand (Demoulin 2019). We were concerned that the interview situation might reinstate more asymmetrical relationships. Therefore, we suggested inversing the roles of interviewer and interviewee during the course of the interview, for example. The relationship of empathy/distance required by an interview situation was transformed because we knew the interviewees, had created confidence and trust, and were engaged in a common project. However, it gave us insights so that topics discussed in the workshop could be developed further.

During these interviews, some statements differed from the positions taken as a group. For example, in the group discussions, young Malian girls had all said that they could only marry a man from their caste. Individually, they described far more subtle positions and strategies and some clearly stated that they rejected this tradition. The group's majority position and the weight of strong personalities had inhibited their individual expression in the workshops. In addition, the fear of being judged by the researcher or the sense of failing to meet their expectations may influence what the young people say in an interview situation. However, these divergent discourses testify above all to a thought in structuring, wavering between different options. This provides some insight into what hesitation and negotiation involve.

Here, the hybridization of methods refers to the idea of keeping together a participative and an objectifying posture. This stance makes critical thinking even more important when it comes to the conditions for collecting research material and the resulting bias. We gathered some very diverse material and conducted a comparative analysis. Thus, the participatory approach included an objectifying dimension, when the young people

worked with analytical distance. Similarly, the objectification took into account the nature of material assembled in a participatory framework.

## 4. How Far Can We Build Together?

These methodological precautions did not prevent tension during the research. In our view, tensions are an integral part of the research process; identifying and discussing them is a useful contribution. Thus, following Rodriguez and Brown's advice, we will now examine the dilemmas of participatory research that includes different people with unequal positions in society and in the research itself (Rodríguez and Brown 2009) (p. 3).

Regardless of the techniques used to build a partnership, social hierarchies and power relations do not disappear in participatory research. Everyone has different issues: reflexivity and operational output for the professionals working with young people; reflexivity, recognition, and remuneration for the young people; scientific production for the researchers. Articulating them inevitably creates imbalances, which vary and are negotiated at different stages of research. Thus, we experimented with a framed partnership (Cornwall and Jewkes 1995), involving different forms of participation which depended on the project phases and the interlocutors.

### 4.1. Defining the Research Protocol

During the preparatory phase, the project belonged to the researchers, particularly the coordinators. We went through the classic tendering process to obtain funding. Seminars involving researchers and professionals were organized before the proposal was drafted. Strict academic norms and rules must be respected when writing a formal proposal, therefore, this exercise was supervised by the researchers alone. They integrated a set of constraints that would limit the project later on.

Furthermore, the team of researchers later expanded compared to the initial core group. As the participatory approach was new to most of us, it generated a great deal of curiosity and raised many questions. Thus, it was clear from the project's launching seminar that researchers and professionals alike were in an experimental situation, one of discovery, trial, and error. This partly helped to justify the place given to the professionals. Thus, the method was reworked during the preliminary discussions, which also meant that vocabulary and questions could be shared. "A cooperative research space" was established in this way (Caillouette and Soussi 2017) and gradually consolidated. In this space, the identities of each one are not merged, but articulated and connected to each other: roles are not interchangeable, but everyone benefits from their respective expertise (Elissalde and Renaud 2010; Belleau 2011). The "discreet mixing that occurs on a small scale" (Bussières and Fontan 2011) challenges the roles, the knowledge, and the professional ethics of each member of the research collective.

However, the young people were not involved in this phase of project building, which was reworked by drawing on three trials: the field trial, the analysis, and the writing and the dissemination of research results.

### 4.2. In the Field

During the workshops, the relationships between professionals, the young people, and researchers took different forms depending on the site and on the age and status of the professionals, who adopted very diverse positions. Some animated the discussion with the researchers, but were fairly discreet to avoid influencing how the young people expressed themselves. Some launched right into the debates like the young people and sometimes adopted a normative position. In this way, one female youth worker stressed the fact that young girls wear short skirts and low necklines at their own risk and are being provocative; one male youth worker uneasy with the religious question, made sure that the topic was not discussed. Indeed, if we consider a generation ranging from 16 to 25 years old, some of the professionals working with young people are young despite their status. They behave like "des grands" (older youth, recognized figures), who are involved in supervising and

educating "les petits" (the youngsters) in the neighborhood community (Salane and Brito 2021). The threshold between young people and professionals is sometimes porous, which led us to work on the relationship between "les grands" and "les petits". We proposed that the professionals could have their own focus group, but we did not manage to organize it. The professionals are actually caught up in the constraints of local youth policies and their organizations' hierarchical systems. Occasional cooperative actions did emerge, but they were based on educational projects, which corresponds to their main shared interest.

A relationship was established with the young people in the workshops, following a protocol for discussions, which they had not helped design. As a result, they took up the proposals in different ways. Some questioned the role of producing videos, for example, and how it fits with group discussions. Some were reticent about any form of expression that might seem academic. The researchers and educators took the criticisms into account to adapt the process, but in most cases they remained in charge of the organization. During the first weekend that brought together all the young people, the research trajectory was more open. They were able to choose the topics to work on. Thus, they rejected certain subjects, suggested others, criticized some discussions, and proposed a new meeting. This took the form of a second working weekend, when they added new topics, leaving some workshops empty and others overinvested. Therefore, the young people were not powerless: they had the power to decide whether or not to attend, to speak or keep quiet, use humor and derision among themselves or with the researchers, and to orient the topics for group work.

During the workshops, a close relationship developed with the young people, engaging the researcher far more than is the case in an interview situation, even when there is empathy. This engagement often appeals to the emotions and involves continuous reflexive work. The researchers supervised discussions and helped the young people put together their videos. As a result, they shifted from their classic observational role. This methodological negotiation between participatory and objectifying posture was disruptive for some. One researcher was worried about losing her methodological rigor and professional specificity, and asked: "Am I turning into an educator? Am I doing research or social work?" The academic work conditions also had an impact. The teacher-researchers were busy with teaching and administrative tasks. They were often involved in different research projects and had little time to think between the workshops, review their notes, listen to recordings, and start identifying avenues for analysis and discussion.

*4.3. Opening the Analytical Box*

Analysis is not a linear process. It starts in the field, with field notes, discussions, occasional assessments, and continues with data processing—interviews, workshop recordings, and reports—and during writing. It corresponded to a specific time of negotiation between objectifying and participatory postures.

The researchers' knowledge of a theoretical corpus and methods form the basis of the analysis. This is one of the key features of their professionalism and guarantees the scientific rigor of their work in relation to their peers. A theoretical corpus and data processing can be heavy and difficult to share with "lay people". Participatory does not remove the confines of the laboratory. Even if formally open, it remains in fact little accessible. Added to which, the issues of academic and intellectual legitimacy crop up again. Indeed, most professionals working with young people consider, for example, that theory is "a field reserved for the world of research that should be developed, but they should not be directly associated with research" (Fontan 2017). The challenge was to establish a dialogue between experiential knowledge, which also has an analytical dimension, and scientific knowledge, which is usually built from theoretical issues and based on rigorous data-processing protocols; and to work on flows, on sharing types of knowledge or shifting from one to another.

First, we organized lectures, bringing together academics and professionals to analyze the outputs of the workshops. We wanted to adopt a participatory posture, but struggled to find a suitable form: the academics came to the lectures with annotated printouts of the workshop reports (several hundred pages per site), whereas the professionals, whose

professional practices do not usually involve that amount of reading, had no time to read the material. The academics presented their preliminary analyses to the professionals in order to amend and improve them. That did not prevent occasional tensions from arising over legitimacy between the youth workers' experiential knowledge and the researchers' interpretations.

Furthermore, we organized monthly thematic lectures during the day at the university, with a classic two- or three-hour program, including formal presentations and discussions. This was to meet the needs expressed by researchers for spaces of scientific dialogue, which corresponded to their auditing standards and working habits. The lectures were open to the professionals, but very few attended and they felt uncomfortable with the program. The possibility of inviting the young people was raised, but rejected because we thought it might reintroduce a hierarchical framework, which we had tried to relax. Nonetheless, reflective work began with the young people during the workshops and was pursued at the time of writing (we will come back to this later).

The transition to the interview analysis was marked by a process of closure, which we had not anticipated. First, given the number of interviews (112), we chose to use a software program to facilitate the analysis, which only the researchers would be able to use. Above all, we were faced with ethical and data protection issues. We had assured the young people that the interview discussions would be anonymized and would not, on any account, be transferred to a third party in that form. We then realized that if we wanted to avoid breaking the pact with the interviewees, we could not share this data with the professionals, who were considered a third party. Some of them found it difficult to accept this brutal closure and the limits set for the shared analytical work. This incident raises the broader issue about the impossibility of having total transparency for research data.

### 4.4. Writing as a Time for Analytical Development: Together and Separately

Although it was difficult to share the work involved in the analysis of the field material, the discussion about the results and their interpretation continued with the young people and professionals until the time of writing. The joint book, which completes the research, took the form of a glossary linked to a website. It presents texts written by the young people and the professionals and researchers, as well as videos and photos. Our editorial choice is based on our desire to show how the flows of thought moved between the words and topics that the young people worked on, between the different forms of expression and analysis, between the territories where these young people live, and between the participants' insights and their different positions with regard to the research. The research showed the extent to which young people's experience is shifting and multifaceted. Appreciating this experience in its entirety and diversity is to acknowledge that not a single dimension can be understood in isolation. A given dimension can only be grasped in relation to other dimensions. Thus, entries in the glossary refer to each other.

Drafting the book was important for analysis and discussion. Some researchers wrote long notes with an analysis of interviews, workshops, and videos. They then proposed short and more accessible written entries for the glossary, while trying to maintain their analytical depth and rigor. The discussions about the texts, especially with the professionals, enhanced the entries. Complementary texts were added by the youth workers or the young people, which broadened the field of analysis through dialogue and feedback loops.

The time dedicated to writing and dissemination of research results was important in terms of the negotiation between participation and objectifying. It raises ethical questions and involves trade-offs regarding important observational material, such as what can be divulged from almost private confidential exchanges and from the tensions or hidden conflicts perceived at different times during the research. In the case of this research, the tensions were significantly limited by the fact that several voices could be expressed. The format for the joint publication was chosen in order to ensure that words and knowledge would be audible without being screened by scientific language, to encourage and discuss knowledge hybridization, and to hybridize the analyses. It is still too early to assess

whether the goals have been achieved for the readers, but the process of writing was an important phase in terms of the collective approach. However, not all the participants reached this stage.

The expectations and systems of validating the research vary for the different partners. Trying to reconcile them is a gamble in itself. Producing a book was a common and mobilizing goal. We also tried to ensure that participants could present and disseminate the work to different target audiences beyond the scientific community, in their various professional and personal spheres. For example, this included presentations with the young people and the professionals in public spaces, and articles in journals at the interface between professional and academic circles. These occasions revealed the disparity between people's expectations. After a conference organized in a university context, but geared to a wider audience, the young people were delighted to have had access to this arena and to have been able to express themselves. In contrast, the researchers regretted that the presentations and discussions remained at the testimonial stage and lacked analytical scope. The research results were also presented in the form of a theatrical performance, based on texts written by the young people and played by young professional actors from distressed neighborhoods. Hearing their texts spoken by other people on stage was a form of recognition for the young people. It also allowed them to distance themselves from their work (Bacqué et al. 2022). A video clip of the first version of the play is visible on the website jeunesdequartier.fr.

Some youth workers were expecting that the research results would bring about transformations in professional practices. They expressed their desire to see researchers working with them to bring about change. Therefore, we organized local presentations of the research, and answered requests to take part in events linked to the research. However, we did not address the issue of extending the mechanisms or tools for action. The researchers' project was not one of "action-research" as a strategy for change to solve problems (Morrissette 2013), although some practitioners in the field wished it had been. The professionals could, of course, use the findings to develop tools for action. Nonetheless, here we want to stress that it was not a major issue for the researchers and, at this stage, they controlled the direction of research.

In parallel, the researchers prepared communications for conferences and started writing articles for scientific journals. This stage represents one method of scientific production. It also means that results can be discussed and validated by peers, a principle that is central to the production of scientific knowledge. It is a necessary passage for researchers which allows them to gain recognition and develop their career. Nonetheless, the process to produce this written work may raise questions in a participatory research framework. Writing according to academic rules and complying with the expected format requires practice. It involves understanding professional norms specific to researchers, which makes co-writing almost impossible. Of course, publications need not be signed by the researchers alone, but could also include the names of the professionals or the young people, in our case. Working together to prepare or discuss the analyses to be presented in publications is also possible, as in our case. The fact remains that in an academic framework, it is ultimately the researcher who takes over, holds the pen, as it were, and speaks because they have the knowledge and codes to make themselves heard in the scientific community.

It is important to note that the professionals' and young people's involvement was not the same throughout the process. The fieldwork was conducted over a short space of time compared to the amount of activities organized (around six months), whereas the analysis took much longer. Researchers are used to working with a long time span, unlike young people and professionals. For the latter, keeping up with a research process over time was difficult when it involved "extra" work: indeed, the time spent was by no means systematically included in work time. Furthermore, the world of professionals working with young people is unstable: some changed organization and could no longer take part in the research in their personal or work time. Others stayed in the same place, but their position in the hierarchy changed, which meant they no longer had the time to take part

in the research once the workshops with the young people were over. The youth workers who participated until the end are, clearly, the ones who were genuinely interested in the project. They were more established and could devote some time to research because it was structurally and materially possible for them to do so. For the young people, the time dedicated to research was in direct competition with the other activities in their busy schedules (work, studies or school, leisure, domestic chores in particular). The trade-offs meant that some continued investing their time, while others simply disappeared because of their trajectories, their changing interest in the research, and the personal relations established with the researchers and the youth workers.

## 5. Conclusions

How does participation affect research? In the case of POP-PART's research, it undeniably contributed to the production of rich and diverse material, which we could not have collected otherwise. Given the plethora of material and the different conditions in which it was gathered, we were able to grasp the heterogeneity and complexity of young people's experience, to identify areas of tension and, above all, to understand how negotiations and compromises work in a constrained framework.

The multiple points of view, the flow of words, and the topics worked on by and with the young people generated a systemic view of youth experience. While the intersectional approaches underlined combinations of class, gender, and race relations, this research revealed the crossover between the latter and generational relations, relations to the territory, and family histories. For example, it demonstrated how much the relationships between "grands" and "petits" structure individual and group trajectories, as well as the social relationships in the neighborhood.

The systematic comparison of different materials, the "cross checks" conducted by the researchers, young people, and professionals working with young people encouraged scientific rigor during the phases in the field and during the analysis. The tension between participatory posture and objectifying posture was uncomfortable, but we sought to keep the two together because they were mutually enriching and stimulating for the researchers. It led them to rethink their role and certain routines, their habits and style of writing, their relationships with their "subjects", and the nature of their requirements in terms of output. Reflexivity was also an integral part of the research. When attempting to examine how research is manufactured, even assuming that some operations involve the researchers' professionalism, failures and errors should not be swept under the carpet, but submitted for criticism. The focus on power relations and their objectification during the research process made us more aware of how situations of inequality influence representations, interactions, and knowledge production.

Lastly, the use of participatory methods helped to develop a participatory ethos within the research team. Many analyses were conducted collectively. They were shared and discussed because each member of the research collective had their own field experience and analysis, which could complement or qualify the proposed interpretations. When the final work was being drafted, the researchers and professionals reread the texts. The young people only read part of the work because the health situation due to COVID-19 prevented us from bringing them together as planned. In the current context, where assessment methods are pushing research production to be more competitive and individualistic, this dynamic deserves special emphasis. However, it requires substantial investment from researchers in terms of time and availability. It is also emotionally demanding because of the relationships involving their research partners. The participative posture heightens ethical issues raised in other types of research. For example, how to stop fieldwork? Can the researcher disappear into their laboratory when they have developed relations with young people and professionals, and expectations have been generated? How far does their responsibility go? What can be said, and how, regarding the elements that emerged during the research process?

**Author Contributions:** All authors contributed equally. All authors have read and agreed to the published version of the manuscript.

**Funding:** This research was funded by Agence Nationale de la Recherche, grant number ANR-17-CE41-0005 and Conseil regional de la recherche en sciences humaines du Canada, Tryspaces project.

**Institutional Review Board Statement:** Our institutions did not require this research to be submitted to an ethics committee.

**Informed Consent Statement:** Informed consent was obtained from all subjects involved in the study.

**Data Availability Statement:** The data presented in this study are available on request from the corresponding author. The data are not publicly available due to privacy.

**Acknowledgments:** This is an adapted translation of «La recherche au défi de la participation. L'expérience de la recherche «Les quartiers populaires au prisme de la jeunesse » » originally published in French by Presses universitaires de France (Sociologie, 2022, 13, 297–315). This translation was prepared by Isis Olivier with support from Conseil regional de la recherche en sciences humaines du Canada, Tryspaces project. Permission was granted by Presses universitaires de France, Jeanne Demoulin and Marie-Hélène Bacqué. The authors would like to thank all the members of the Pop-Part Team for their contribution to the research.

**Conflicts of Interest:** There are no conflict of interest to declare.

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
