# Peer review of "Research and the Challenge of Participation—The Experience of the Research Project “Distressed Neighborhoods through the Prism of Youth” (ANR Pop-Part)"

_socsci, doi:10.3390/socsci12070415_

Round 1

Reviewer 1 Report

Dear author(s),

After carefully reading your study, here you can find some comments that I hope will help improve your work: 

Point 1: Research gap and research questions need to be addressed in a clear and punctual way.

1.1. In this study, different questions are stated throughout the whole manuscript (see lines 58-60, 62-63, and 183). This confuses the reader about the objectives and directions of the research;

1.2. In the same way, the need for this research as a valuable scientific contribution to participatory research theories is emphasized. However, without specifying the research gap. For instance, in the introduction, the research gap could be specified while transitioning between the brief description of the participatory research and the specific research conducted in this study (i.e., lines 21-22).  

Specifying the research gap in a more detailed way would allow for stating the specific research objective. For example, saying that this study supports the understanding of how much PAR contributes to scientific production and what the limitations are (lines 23-25) becomes redundant considering that there is extensive literature on the subject. Instead, specifying the nuances of the context or conditions under which this process takes place could better delimit the research.

Point 2: the structure of the manuscript. 

2.1. According to the specificity of this manuscript, it is recommended to follow the structure given in the journal guidelines: 1. introduction; 2. literature review; 3. material and methods; 4. results and discussion; 5. conclusion;

2.2. At the same time, it is also recommended to improve the structure of the abstract and the information provided in it. It might help to consider the following structure: problem definition, research question, applied theory and methodology, main findings and conclusions (and research limitations).

Point 3: Literature review and results’ discussion

 3.1. In the literature review section, it is suggested to frame the information on participatory research into an initial paragraph describing the reasons, the different approaches and the terminology associated with it. It is also suggested to include the information in the footnotes (e.g., footnote no. 3); 

3.2. When describing the terminology associated with participatory research, it is highly recommended to emphasize also the difference between action research and participatory action research (the first one mentioned in lines 83-84) due to the different approaches and social implications they have;

3.3. In order to ensure the clarity of the results, the different stages conducted for the various groups involved (which appear to be different) need to be clarified by highlighting the scientific rigour of the choices of methods, and participatory tools used; 

3.4. The literature should be expanded to provide more scientific information about the participatory methods and instruments used, but especially about the involvement of non-researchers in carrying out this research. In this case, it might be useful to refer not only to theories but to find 'good practices' where this approach has been used on an empirical level. This might give more consistency to the kind of bias that this approach might bring to research, as well as largely justify the choice of sharing the project approach with a wider audience in the different stages of idealisation, structuring, conducting the project and analysing the results. This would allow for a better contextualization of results (see lines 287-289 and 475-481, for example);

3.5. It is strongly suggested to provide more information on the crucial concepts for this research. For example, it is necessary to define and contextualise the work of POP-PART, of which very little is mentioned. it is also necessary to provide a definition for terms such as "youth workers" (lines 443,452,473) which is of ambiguous interpretation. It is also suggested to give a definition to specific social phenomena mentioned, such as 'gentrification' (line 315) in order to ensure broad accessibility for reading and understanding the research. The same happens for “popular education methods”(righe 226-227 e 290). What are? How are they characterized for? Some examples? A definition is also needed for "disadvantaged youth" (lines 191-192). It is also suggested to specify the reasons why some statements are made, for example: "the participatory research [...] is now attracting renewed interest" (lines 15-16), one may ask: why?; 

3.6. It is suggested to specify some contingencies and their impact on the research, e.g. "the young people only read part of the work because the health situation prevented us from bringing them together as planned" (e.g. lines 685-687). What is this health situation?

3.7. There is no discussion section in this manuscript.  The discussion is essential to enable scientific analysis and commentary of the results, starting with the literature review that the authors identify relevant for this study;

3.8. According to the features of this study, it might be useful to present the results and discuss them simultaneously. Therefore, to have a 'results and discussion' section.

4. It is strongly recommend pay attention to ethical aspects of this research.

4.1. First of all, the research involves minors (the target group mentioned in footnote no. 7 is 15 to 33 years old). Therefore, it is recommended to anonymising the data by assigning a code to each participant. It might be helpful to avoid explicit reference to the person involved (see lines 338-339);  

4.2. Attention should be paid to questions concerning the remuneration/ financial commitment of the people involved (a matter to be consulted with specialists, as the research ethics committee); 

5. Conclusions and references

5.1. The conclusions need to be better articulated, summarising what was discovered through this study, according with the research gap, the research questions and the proposed methodology;

5.2. Also, the concluding part of this study presents some of the text from the template used for the article (lines 689 - 717);

5.3. when citing part of the text of a third-party contribution, it is strongly suggested to state the exact page number or pages within the citation (e.g., lines 17-18, De Roubaix & De Cronik, 2013, p.XXX);

5.4. it is suggested to put the references at the end of the text (e.g. line 21);

5.5. it is suggested to remove all the footnotes (which are not allowed by the journal) and to convert them in standard text of the manuscript; 

5.6. it is suggested to provide the DOI (when available) for all the references used in the study; 

5.7. overall, it is also suggested to provide more international literature for this study.

Dear authors,

The editing of the English language is recommended. This could help to improve sections of the text that require editing or further elaboration.

Reviewer 2 Report

I liked this paper.  The only significant problems I found were with the last four paragraphs.  While the rest of the paper presents the complexities and ambiguities of a youth and practitioner involved participatory action research project, this part of the paper sounds like it was written by a different author.  The focus in this last section is on submission requirements for quantitative papers that use large, publicly available data sets.  Data for a qualitative, participatory paper like this one cannot easily be made available to either reviewers or the audience.  While the conditions presented in these last four paragraphs are required by an increasing number of journals, I'd think these authors would make a comment about the difficulty of making the sort of data described in this paper available.   Indeed, I'd prefer to see a critical comment about the inappropriateness of those data availability norms for many qualitative research projects.

I found the following typoes:

p. 1, l 27, insert "the" before Paris

p. 5, l 201, separate power and relations

p. 7, l 306, "our" views rather than "out" views

p. 9, l 376, should there be an "s" at the end of arrondissement?  In English an "s" is needed for a plural

p. 13, l 595, "reconcile" not "reconciling"

p. 15, l 705, I think "implies" is a better word choice than "implicates"

No problems...I've listed some typoes that might be hard to find.

Author Response

The 4 last paragraphs should not have appeared in the manuscript. This was an error when using the template. We have removed them.

We have corrected the typos, thank you!

Reviewer 3 Report

Thank you for asking me to review this article. It is clear, well-written and easy to follow. I found the work to be interesting and of high quality. Participation is a big topic to take on and you have done a great job. There is relatively little work describing the 'nuts and bolts' of methodologies in practice and how both researchers and participants are impacted. As well, enough context of the actual research is given to make the descriptions and discussion very useful. I see no need for any substantive changes to the work. 

Just a couple of comments. In line 306, I am thinking that it should read, "our views" and in line 435, the word might be "extent" rather than "extend".  That is it. 

Thanks again!

Only one or two minor changes are needed. See comments above.

Author Response

We have corrected those typos, thank you!